# Phage Display to Augment Biomaterial Function

**DOI:** 10.3390/ijms21175994

**Published:** 2020-08-20

**Authors:** Thomas A. Davidson, Samantha J. McGoldrick, David H. Kohn

**Affiliations:** 1Department of Biomedical Engineering, University of Michigan, Ann Arbor, MI 48109, USA; tadavids@umich.edu (T.A.D.); smcgoldr@umich.edu (S.J.M.); 2Department of Biologic and Material Sciences, University of Michigan, Ann Arbor, MI 48109, USA

**Keywords:** phage display, tissue engineering, biomaterials, dual-functioning peptides, peptides

## Abstract

Biomaterial design relies on controlling interactions between materials and their biological environments to modulate the functions of proteins, cells, and tissues. Phage display is a powerful tool that can be used to discover peptide sequences with high affinity for a desired target. When incorporated into biomaterial design, peptides identified via phage display can functionalize material surfaces to control the interaction between a biomaterial and its local microenvironment. A targeting peptide has high specificity for a given target, allowing for homing a specific protein, cell, tissue, or other material to a biomaterial. A functional peptide has an affinity for a given protein, cell, or tissue, but also modulates its target’s activity upon binding. Biomaterials can be further enhanced using a combination of targeting and/or functional peptides to create dual-functional peptides for bridging two targets or modulating the behavior of a specific protein or cell. This review will examine current and future applications of phage display for the augmentation of biomaterials.

## 1. Introduction

Phage display is a selection technique used to identify amino acid sequences based on their affinity for a given substrate. Due to the random nature of phage libraries, phage display identifies known, but also new, non-obvious sequences that could not have been identified via protein engineering strategies. In general, the peptide sequences identified via phage display can be divided into targeting peptides and functional peptides. Targeting peptides have high affinity for a specific substrate, cell, or protein. Functional peptides modulate cell or protein activity upon binding to that cell or protein. Targeting peptides, functional peptides, or a combination of the two, can be implemented in biomaterial design for functionalization of materials with the goal of increasing the ability of the materials to instruct cells.

Biomaterials require intelligent design in order to manipulate interactions between the biomaterials and the surrounding microenvironment. Phage display aids in the discovery of peptides with high affinity and specificity for a target, which can be used to guide interactions between the biomaterial and its environment. Functional molecules can elicit specific cellular behaviors upon interaction with the designed biomaterial. Targeting molecules can improve homing of biomaterials to specific cell types or add biological cues to a material that cells are unable to recognize. Dual-functional peptides add even greater versatility to biomaterials. For example, combining a functional molecule and targeting molecule within a biomaterial can elicit a desired behavior from a specific cell type or cell subpopulation. Alternatively, combining two targeting molecules within a biomaterial can bridge two materials or recruit a specific cell type to a given material.

This review aims to show the potential for the use of peptides identified via phage display in biomaterial engineering. First, we outline the general phage display technique and discuss technological limitations along with future advancements. Then, we highlight targeting versus functional peptides identified via phage display and describe how each can be utilized for the functionalization of biomaterials. Lastly, we describe the concept of dual-functionalized peptides for a modular approach to biomaterial design. In all, the reader should come away with an understanding of how phage display identifies previously unknown peptides with targeting or functional capabilities that can be used to augment and customize biomaterials for a vast array of biomedical applications.

### 1.1. Phage Display

Bacteriophages are viruses utilized to both encode and display a library of amino acid sequences. The sequence is encoded within the bacteriophage DNA via a mutation insertion sequence and is ultimately displayed on the outer coat of the bacteriophage which provides simultaneous genotypic and phenotypic expression of the peptide.

Two of the most common phages are the M13 and T7 bacteriophages. Phage display was first developed using the M13 bacteriophage, and the M13 bacteriophage is well characterized and easily engineered to express a variety of peptides and proteins [1,2]. Most commonly, the filamentous M13 bacteriophage is used to display up to 5 copies of an encoded peptide sequence (6–45 amino acids) on its pIII minor coat protein [3,4]. The M13 phage is replicated in *Escherichia coli*, but the release of the phage from the bacteria requires cooperation with the host secretory pathways [3]. This requirement limits the peptides that are actually displayed on the M13 phage to those that work within the host secretory pathways, despite the theoretical peptide diversity of the phage library [5,6]. To bypass this limitation, the T7 bacteriophage was adapted for phage display [5,6]. T7 bacteriophage also replicates using *E. coli*, but the phage is released upon lysis of the host cells [5,6]. This allows for phage display of all potential peptides in a given library, rather than only those that can be secreted by host bacteria [5,6]. T7 bacteriophage displays peptide sequences (40–50 amino acids) or proteins (900–1200 amino acids) on protein 10, with up to 415 copies of smaller peptides and one or fewer copies of large proteins [5].

Phage libraries are created by introducing a mutation insertion sequence into the bacteriophage genome. This establishes a large set of phages (10^9^–10^12^) that display different amino acid sequences based on the general design of the insertion sequence. Depending on the design of the insertion sequence, the expressed peptides can be linear or cyclic and can range in size and composition.

Typically, phage display with a random peptide library consists of multiple rounds of biopanning, in which a phage library is exposed to the substrate of interest and the highest affinity binders are recovered (Figure 1) [7]. Care must be taken to wash the substrate thoroughly in each panning step to ensure that non-bound or weakly bound phages are removed. Once bound phages are eluted off the substrate, they are used to infect bacteria for amplification of the phages. Colonies can be harvested, and the phages can be isolated for another round of panning and/or for sequencing. After multiple rounds of panning, the highest affinity binders can be identified, characterized, and tested for their specificity to the desired substrate in order to identify the top candidates.

The general phage display technique can be used to identify both functional peptide sequences and targeting peptide sequences. Phage display is ideally suited for identifying targeting peptide sequences through repeated rounds of panning against a specific substrate to isolate the highest affinity binders. However, one can utilize phage display to identify functional peptide sequences that elicit a change in cellular or protein behavior upon binding by using functional measures in combination with binding affinity.

### 1.2. In Vivo Phage Display

While phage display is largely performed in vitro, in vivo phage display expands the potential applications for peptide discovery and biomaterial customization. In vivo phage display involves local or systemic injection of the phage library into an animal. Then, both target and control tissues are harvested so that bound phages can be eluted, amplified, and analyzed. In vivo phage display provides the combination of positive and negative selection since phages can be identified for their binding to the target tissue and lack of binding to the control tissue in the same animal. In vivo phage display relies on high specificity of target-bound phages and efficient amplification of phages.

The first demonstration of in vivo phage display for organ-selective targeting identified the peptide sequence CLSSRLDAC for its affinity to brain tissue [8]. Advancement in technology has allowed for the use of in vivo phage display for the identification of peptides with higher selectivity for a specific tissue type. For example, one group sought to increase selectivity for xenografts of human prostate cancer in mice by performing multiple rounds of negative selection prior to in vivo phage display for a prostate cancer tissue homing peptide [9]. In each of three rounds of negative selection in mice and one subsequent round of negative selection in cynomolgus monkeys, the phage library was injected into the bloodstream and non-bound phages were harvested and identified [9]. This step eliminated phages with known in vivo targets from the library, so that in vivo phage display could be used to identify highly selective prostate cancer tissue homing peptides [9].

In addition, in vivo phage display advances have provided means for identifying peptides with greater targeting capacity for pathological tissues, even without known targets. For instance, in vivo phage display identified the peptide QPIHPNNM for its affinity and selectivity for adenomas over normal colon tissue [10]. Endoscopic monitoring of morphological changes in adenomas is used to manage colorectal cancer progression, but it could be improved with QPIHIPNNM-based targeting and labeling of adenomas over the current standard of qualitative observations (Figure 2) [10]. In vivo peptide phage display identified CSPGAKVRC (nicknamed UNO) for its affinity to tumor-associated macrophages (TAMs) [11]. UNO interacts with TAMs through CD206, and UNO can target solid tumors of different cancer types, making UNO a viable candidate for targeting *CD206^+^* tumors for diagnosis, imaging, or drug delivery [11]. In vivo phage display was also used to identify the peptide CAQK for its affinity to brain tissue affected by a penetrating brain injury that allowed phages to cross the blood-brain barrier (BBB) [12]. In cases where the BBB has been disrupted due to injury, CAQK could be used to guide imaging or therapeutic molecules to the injury site, bypassing invasive delivery measures [12]. In both of these examples, in vivo phage display was used to identify peptides that could target specific types of pathological tissue to improve diagnostic and therapeutic strategies.

### 1.3. Processing Phage Display Data

After multiple rounds of biopanning and amplification, the number of potential peptide sequences will be reduced from ~10^9^ to 10^2^–10^3^. The remaining candidates must be further analyzed to determine the top candidates based on binding strength and selectivity for the desired target (Figure 3).

When phages are selected after biopanning and enrichment, they are sequenced in order to identify the peptide sequences encoded in the identified phages. Typically, Sanger sequencing is used to characterize <100 peptide sequences from multiple rounds of biopanning in order to find consensus sequences [13]. However, next-generation sequencing (NGS) offers the ability to sequence larger portions of the peptide library (e.g., Illumina sequencing can read over 10^7^ sequences) [13]. Increased sample sizes for sequencing can lead to identification of peptide sequences that may have been under-represented by traditional biopanning and Sanger sequencing, due to poor amplification [13,14]. NGS identifies longer sequences than traditional Sanger sequencing [15]. NGS can also identify peptides based on metrics other than binding affinity, giving researchers more flexibility in utilizing peptides for biomaterial augmentation [14]. For example, one group used in vivo landscape phage display library in combination with NGS to identify a number of 3-mer consensus sequences or elementary binding units (EBUs) based on peptide avidity, rather than peptide affinity [16]. The EBU motifs were categorized as unique to a single tissue type, common to multiple, similar tissue-types, or common to all tissue types [16]. This information was then used to discover a subset of EBUs unique to a tumor microenvironment, allowing for the directed evolution of the landscape phage display library for the identification of a tumor-specific peptide sequence [16]. In all, NGS is valuable for analyzing large peptide libraries, reducing costs and time with increasing magnitudes of sequences [15].

Once the number of selected peptides has been narrowed down by sequencing and identification of consensus sequences, binding assays are employed to quantify the peptides’ binding strength and selectivity for the desired target. A combination of competitive and non-competitive binding assays can quantify the association and dissociation constants of the peptides relative to their intended targets [17]. The enzyme-linked immunosorbent assay (ELISA) is a common tool utilized for the quantification of binding affinities of peptides to known targets and can be easily customized for the desired targets [18,19,20,21,22,23].

In the cases of cell or tissue targets, human biopsies can be used to assess peptide binding to validate the sequences identified via panning against in vitro or animal models of in vivo targets. Due to the variable nature of human biopsies, it is important to verify that the binding affinities and selectivities of peptides identified via phage display can translate to ex vivo human tissues. For example, a microarray of human prostate cancer tissues was used to validate the binding of the peptide PC204 discovered via phage display against a prostate cancer cell line [19]. It was found that 74% of the tumor specimens exhibited positive binding to PC204, demonstrating that PC204 could recognize the majority of clinical specimens even though the cellular recognition motif had yet to be identified [19]. Additionally, after identifying multiple cervical cancer-specific targeting peptides (CSPs) via in vivo phage display, a microarray of human cervical cancer tissues was used to confirm selectivity for cervical cancer tissue over normal cervical tissue [24]. In this example, two different CSPs were identified for their selectivity to different presentations of cervical cancer: CSP-GD was found to selectively bind to adenocarcinoma (82% of cases showed positive binding) over squamous cell carcinoma (8%) and normal cervical tissue (6%), while CSP-KQ was found to selectively bind to squamous cell carcinoma (80% of cases showed positive binding) over adenocarcinoma (7%) and normal cervical tissue (1%) [24].

For phage display against cells or tissues, the localization of the phage may provide additional information on the target and/or function of the displayed peptide. The phage could be sorted as membrane-bound or internalized by recovering the phage before and after cell lysis [25]. Alternatively, imaging fluorescently tagged peptides can demonstrate whether or not the peptide is internalized by the target cell [22,26]. qPhage was developed for quantifying the internalization of phages with RT-PCR of cells after exposure to a phage library [15]. qPhage requires a shorter incubation time than immunostaining and is more sensitive for the detection of internalized phages [15]. Another approach to identifying peptides with intracellular targets or functions is to use internalizing phages, rather than relying on sorting peptides by their internalization activity. An internalizing phage library (iPhage) is generated by adding a cell-penetrating peptide (CPP) to the coat protein to allow for simultaneous expression of the CPP and peptide library [27]. The use of iPhage expands the application of phage display to internal cellular targets, such as organelles and intracellular signaling pathways, which are difficult to target [27]. Designing biomaterial systems based on peptides identified via the iPhage system would still require the incorporation of an internalization mechanism or CPP since the peptides themselves are not necessarily internalized [27]. Taken together, phage display can be used to identify peptides that are naturally internalized by cells or have distinct intracellular targets, both of which are key for expanding the identification of functional peptides for biomaterials.

Based on the desired biomaterial application of the peptides identified via phage display, a combination of processing techniques may be required to verify a peptide’s binding affinity and selectivity for a given target. For material targets, post-processing focuses on quantifying a peptide’s affinity and demonstrating the peptide’s selectivity for the material. For cell- and tissue-based targets, processing focuses on quantifying a peptide’s affinity for a given target, demonstrating the peptide’s selectivity for that target, and probing the interaction mechanism between the peptide and target. Post-processing techniques aim to bring the pool of candidate peptides down from 100+ sequences to <10 sequences.

### 1.4. Limitations and Improvements to Phage Display Technology

Though analysis of candidate sequences operates on the assumption that libraries are random and unbiased, there are limitations that conflict with that assumption. Due to a preference for certain amino acids and early termination due to stop codons, libraries do not contain all possible peptide sequences and do not necessarily contain equal numbers of the same peptide sequences [28,29]. In addition, due to over- or under-representation of certain sequences, the isolation of the true highest affinity binders can be compromised [28,29]. Therefore, care must be taken to either create a truly unbiased, random, amino acid library or to recognize inherent limitations to phage display technology and subsequent analysis. For instance, one group used ‘deep panning’ to sequence a random peptide library (7-mer sequence of NNK codon, where N = any nucleotide and K = G or T) and found over-representation of the UAG amber stop codon and a bias for glutamine [28]. In a subsequent experiment, neither of these outcomes occurred when the bacteria used to synthesize the peptide library were altered to suppress the translation of UAG, and the oligonucleotide mixtures were calibrated via mass spectroscopy [29]. As next-generation sequencing becomes more widely used, phage libraries should be screened to ensure true randomness and lack of bias.

Another technical limitation of phage display is the identification of false positives due to peptides being recovered after binding to non-targets. However, sequences that bind to components other than the targets can be collected. For example, SAROTUP (Scanner And Reporter Of Target-Unrelated Peptides) is searchable online database for peptides known to bind to unintended targets, such as plastic or streptavidin [30,31]. Additionally, a microfluidic phage selection (MiPS) system, where the labor-intensive steps of incubation, washing, cell lysis, and lysate collection are fully automated, and human error, identification of non-specific binders due to improper washing, and the need for large numbers of adherent cells are reduced [32]. With the MiPS system, smaller, well-characterized populations of cells or even biopsies can be effectively used for phage display, expanding the potential use of phage display into the area of personalized medicine [32].

Phage display can be compromised by the prevalence of parasitic clones or phages that are more easily amplified in bacteria. Parasitic clones become over-represented in the phage population, resulting in bias toward identification of the parasitic clones over other, potentially stronger binders. To demonstrate the negative effects of parasitic clones, Illumina deep-sequencing technology was used to analyze a portion of a phage display library before and after amplification in bacteria [13]. There was a significant decrease in diversity after a single round of amplification in bacteria that could not be observed with smaller-scale Sanger sequencing [13]. Illumina NGS could then be used to identify parasitic clones from the commonly used Ph.D.-7 library and it was shown that performing amplification in an emulsion could also reduce amplification bias [33].

A significant limitation of phage display is the lack of post-translational modifications within the expressed peptides on the phage surface. The incorporation of post-translational modifications during phage display post-processing increases the naturally limited diversity of peptides made from 20 amino acids. Glycosylation, cyclized peptides, and phosphorylation may all affect the potency or specificity of the peptide ligand. For example, phosphorylation of two serine residues of the peptide VTKHLNQISQSY (VTK), identified for binding to bone-like mineral, a carbonated apatite with a similar composition to the inorganic phase of bone, showed a 10-fold increase in adsorption compared to the unphosphorylated control [34]. Inclusions of these post-translational modifications into phage display libraries could lead to identification of ligands that would otherwise not exist in purely linear peptide libraries.

While the general philosophy of phage display has remained the same over time, advances in automation of labor-intensive steps and integration with big data have enhanced computational analysis. NGS provides more data on the interactions between the peptide library and the desired substrate, making the application of the phage display process more flexible and akin to big data [15]. With NGS, one could identify peptides that may not necessarily be the highest affinity binders, but may be more infective, better at amplification, or have higher avidity [14,16]. Additionally, NGS can help identify consensus sequences faster than traditional methods and can help identify longer sequences than that generally identified by traditional Sanger sequencing [15]. For example, one group used NGS after each round of traditional biopanning with a commercial phage display library against KS483 cells during different stages of osteoblastic differentiation [35]. They demonstrated that a single round of selection combined with NGS sufficiently enriched the phage population, suggesting that performing the first round of selection in duplicate or triplicate and sequencing with NGS may be a preferable method for identifying high-affinity peptides via phage display [35]. Though costly, NGS can reduce labor time and overall costs when used to identify large numbers of sequences (Figure 4) [15].

## 2. Phage Display and Biomaterials

Biomaterials are materials designed to interface or interact with cells, tissues, or organs. Often, biomaterials are designed to mimic nature in order to elicit a specific biological response. However, not all biomaterials are made of natural or naturally inspired materials. Thus, functionalization of the material needs to be included in the design to enable direct interaction between the biomaterial and organism.

Phage display provides an opportunity to identify peptides for functionalization of biomaterials. In general, peptides are better than proteins for functionalization because they are smaller, cheaper and easier to synthesize, less likely to lose function during conjugation, and more resilient to changes in the microenvironment, such as a change in pH. Depending on the intended use of a biomaterial, the design process could involve phage display for the identification of different types of peptides. Targeting peptides can be used to functionalize a biomaterial based on the peptides’ affinity for a specific substrate or cell type. Functional peptides can be used to functionalize a biomaterial based on the peptides’ ability to elicit a desired cellular response. A dual-functional peptide can be used to functionalize a biomaterial for joining substrates and/or cell types or for eliciting a desired function from a specific cell type.

### 2.1. Functional Molecules

Functional peptides modulate protein activity or cellular behavior when bound to a protein or cell. When conjugated to a biomaterial, these peptides give the biomaterial greater control over the microenvironment. The presence of functional peptides on a biomaterial can prompt the migration of desired cell populations to the material surface or differentiate stem cells down a specific lineage to induce tissue-specific regeneration. On the other hand, cell function antagonists can prevent less desirable responses, such as pathological calcification.

#### 2.1.1. Adhesion and Migration

Many biomaterials take inspiration from extracellular matrix (ECM) proteins in order to add cell recognition motifs to enhance interactions with cells they would not recognize or mount an immune response against. For example, RGD is derived from the active domain of fibronectin and is one of the most common peptide sequences for promotion of adhesion via integrins. However, phage display reveals new amino acid sequences that also interact with ECM proteins responsible for adhesion and/or migration. For example, phage display was used to identify the peptide sequence DPIYALSWSGMA (DPI) that increases adhesion strength and spreading of human marrow stromal cells (MSCs) [36]. While the mechanism of interaction has not been fully characterized, DPI interacts with MSCs via integrins [37]. DPI has been used to functionalize a biomaterial surface leading to increased specificity of MSC attachment over fibroblasts and osteoblasts and increased magnitude of cell adhesion strength [36]. Another group used phage display to identify the peptide E7 for its affinity to bone marrow-derived MSCs [38]. Then, to recruit MSCs for the regeneration of irregularly shaped cartilage defects, particles of E7 and demineralized bone matrix (DBM) were added to a chitosan (CS) hydrogel [39]. The DBM-E7/CS scaffold improved chondrogenic differentiation of MSCs in vitro and yielded promising cartilage-like structures in vivo [39]. This strategy uses phage display to identify peptides for surface engineering of biomaterials to enable the recruitment of a local MSC population in vivo to aid in tissue regeneration by bypassing the use of exogenous cells.

#### 2.1.2. Differentiation

For the use of biomaterials in tissue engineering, one strategy is to design a biomaterial to present cues to local cell populations to initiate differentiation. Phage display can identify peptide sequences that have an affinity for different stem cell populations, with the goal of targeting subpopulations with greater differentiation. This approach requires panning against a well-characterized cell population, identifying the most promising binders, and evaluating the effects of the most promising binders on differentiation. For example, following phage display against a neuronal stem cell population, membrane-bound or internalized peptide sequences were separately characterized for effects on differentiation [25]. Peptides that directly interacted with membrane-bound proteins or peptides that were internalized by neuronal stem cells (NSCs) were deemed more likely to modulate differentiation. When used to functionalize scaffolds designed to mimic the native stiffness of brain tissue, the identified sequences enhanced NSC binding and differentiation into three mature neural phenotypes [25]. This outcome demonstrates the potential use for peptides in surface engineering of biomaterials to direct local stem cell populations toward specific cell lineages for tissue engineering and regeneration.

As mentioned previously, another approach to facilitating the differentiation of a local stem cell population would use peptides identified via phage display to recruit a stem cell population. Simply, one could use the peptide to recruit a stem cell population to a biomaterial containing known differentiation cues. This has been demonstrated in a number of studies aimed at identifying peptides specific to marrow stromal cells, with the ultimate goal to functionalize bone-promoting biomaterials for the recruitment of MSCs [36,37,38,39,40]. Alternatively, one could explore if and how the interaction of that peptide with the stem cell population impacts differentiation. For example, the peptide DPI, identified for its specificity to MSCs, was conjugated to an apatite-specific peptide and anchored to a biomimetic apatite substrate [36]. When MSCs were cultured on this peptide-coated apatite substrate, markers of both early- and late-stage osteogenic differentiation were upregulated, as compared with an uncoated apatite substrate [36]. This indicated that DPI may play a part in both the recruitment and osteogenic differentiation of MSCs [36]. Using either approach, the identified peptides could then be displayed on a biomaterial to provide cues for differentiation to a specific population of undifferentiated or partially differentiated stem cells in an effort to regenerate local, impaired tissue.

#### 2.1.3. Inhibition of Cell Function

Phage display can also identify peptide sequences that interfere with known targets and unwanted functions. This strategy requires knowledge of the mechanism of action governing the unwanted function since specific cells or proteins must be targeted. Alternatively, this strategy can elucidate mechanisms of action based on interactions between identified peptide sequences and specific cells or proteins. One group compiled a list of peptides identified via phage display for their affinity to a range of known enzymes [17]. They found that 13 of the 17 tested peptides inhibited the respective enzyme’s activity through binding to functional sites rather than binding to random locations on the enzyme’s surface [17]. In this case, the investigators were able to use phage display to identify peptides with high affinities for enzymes, but they needed to verify the inhibitory activity levels of the peptides [17].

Phage display can also be used to identify peptide sequences for their affinity to a specific structure, composition, or orientation on a substrate, inhibiting transformation upon binding. For example, the peptide sequence VTKHLNQISQSY was identified via phage display for its affinity to bone-like mineral [34]. Upon post-translational phosphorylation of the serine residues, the phosphorylated variant, pVTK, inhibited mineralization of osteoblasts [34]. It has been hypothesized that pVTK binds to early mineral deposits and inhibits crystal growth [34]. Therefore, biomaterials can incorporate pVTK for the treatment of ectopic mineralization, taking advantage of the functional nature of pVTK for both targeting unwanted mineral in soft tissues and inhibiting further mineralization.

Together, these two examples establish how phage display is used to identify peptides that modify protein or cell functions based on known targets or known conformations. These examples also illustrate how phage display can identify peptides with unanticipated effects on cell functions. In either case, the peptides identified can be used to functionalize materials to either promote or inhibit cellular responses to implanted or injected biomaterials.

## 3. Targeting Molecules

Targeting peptides have specific affinity for a given material or cell type. When conjugated to a biomaterial, these peptides confer the same affinity to the biomaterial. Peptide engineering of biomaterials surfaces thus allows for intelligent biomaterial design to home to specific targets.

### 3.1. Locate Specific Cells or Tissue Types

Phage display is suited for identifying peptides with high affinity for specific groups of cells and cell receptors, proteins, and compositional elements of the tissue extracellular matrix. This strategy relies on the isolation and characterization of a given cell type or tissue feature. Alternatively, characterization of the cell population or tissue can be done after high-affinity peptides have been identified. For instance, in vivo phage display [41] identified cells within the vascular system having different ‘zip codes’, or sets of receptors [41]. This strategy separates cell populations based on peptides that target different zip codes, allowing for subsequent characterization of these distinct cell populations [41].

#### 3.1.1. Imaging and Diagnosis

The conjugation of imaging molecules to peptides with high affinity for specific cells or tissue constituents is a simple, modifiable approach for improved imaging of pathologies. This strategy can be employed to target areas of the body that are traditionally difficult to target. For instance, ovarian cancer is difficult to detect at early stages because it generally does not present with symptoms, but early detection of ovarian cancer leads to higher survival rates [42]. Phage display against human ovarian cancer cells was used to identify the peptide sequence pJ18, which could guide radiolabels to ovarian cancer cells for detection via single photon emission computed tomography (SPECT)) and computed tomography (CT) imaging [42]. The vast potential for phage display to identify pathological targets is particularly suited for conjugation with imaging molecules for more accurate imaging and diagnosis of almost any pathology.

Peptides with high affinity for specific proteins or cells characteristic of a disease state can be utilized in the design of diagnostics. Many diseases rely on RT-PCR or antibody immunoassays, which require expensive equipment and are not suitable for efficient clinical testing. Phage display technology can be leveraged to identify peptides that will bind to proteins characteristic of certain disease states. Then, those peptides can be utilized in the design of microfluidic devices or biosensors to diagnose diseases based on small samples of bodily fluid. An electrochemical biosensor was designed for the detection of norovirus by using phage display to identify a peptide sequence with affinity for a recombinant norovirus capsid protein [18]. Then, the identified peptide was conjugated to a gold electrode to enable the detection of positive binding between the peptide and norovirus protein via quartz crystal microbalance (QCM) or electrochemical impedance spectroscopy (EIS) [20]. This example demonstrates the use of phage display to identify peptides for improved diagnosis, for example, by increasing the speed of clinical, bedside testing through the use of high-affinity peptides in microfluidic devices.

#### 3.1.2. Targeted Drug Delivery

Phage display is also suited for identifying peptide sequences that can be added onto drug delivery systems for spatial and biochemical targeting. Peptides can be directly conjugated to the drug in cases where conjugation would not affect the drug’s activity and protection of the drug is not required. For instance, poor prognosis is associated with prostate cancer when the cancer is resistant to castration and/or when it metastasizes, but, at this point, treatment often needs to be halted due to poor systemic effects of currently available drugs [9]. One group used in vivo phage display to discover a peptide, C-TGTPARQ-C (LN1), with high affinity and selectivity for prostate cancer tissue [9]. They conjugated LN1 to the peptide _D_(KLAKLAK)_2_, which is known to induce apoptosis, to form a targeted anti-prostate cancer cell therapeutic, LN1-KLA [9]. They demonstrated that LN1-KLA inhibited prostate cancer cell proliferation in vitro and inhibited tumor growth in vivo [9]. This example demonstrates how homing peptides discovered via phage display can be directly conjugated to drugs for targeted delivery to a desired cell or tissue type.

Peptides can also be added to different drug delivery systems, such as nanoparticles, liposomes, or micelles, to guide the systems without directly interacting with the delivered drug. Accurate targeting by drug delivery systems is critical to improve the efficiency of drug dosing and reduce off-target effects. For example, when combating antibiotic-resistant bacteria, such as *Staphylococcus aureus*, one must rely on strong antibiotics that have negative side effects due to the necessity for systemic administration [43]. Therefore, it would be desirable to design a system to target the delivery of antibiotics to the *S. aureus* bacteria and protect healthy cells [43]. One group used phage display to identify the peptide CARG for its affinity to *S. aureus* bacteria and demonstrated its in vivo homing and binding ability to infected lung tissue (Figure 5) [43]. A CARG-coated porous silicon nanoparticle (pSiNP) drug delivery system was also designed for vancomycin [43]. CARG is capable of guiding the pSiNPs to in vivo sites infected by *S. aureus* [43]. Another application for peptides identified via phage display is targeted delivery of chemotherapy drugs. The peptide SP204 was identified for its high affinity to a number of human prostate cancer cell lines, with verification against a microarray of human prostate cancer tissues [19]. SP204 conjugated to liposomal nanoparticles for targeted delivery of doxorubicin against human prostate cancer cell xenografts in vivo shows no significant changes in body weight with administration of the drug-containing SP204-conjugated liposomes, which indicated no significant off-target toxicity upon systemic delivery of the targeted nanoparticles [19]. These are both examples of how peptides identified via phage display can be used for homing drug delivery systems to specific types of pathological tissues, regardless of a known tissue-specific target.

### 3.2. Material Specific Binding

Phage display is also useful for detecting peptides with affinity toward inorganic and organic synthetic substrates. This technique has been used to isolate peptide sequences with attraction toward titanium [44], silver [45], palladium [46], platinum [47], and carbon nanotubes [32,48]. More recently it has been used to find sequences with specific affinity for specific synthetic substrate chemistries.

#### 3.2.1. Silicon

Silicon is a commonly used semiconductor material in both micro- and nano-electronics. Attaching biomolecules such as peptides to act as molecular crosslinkers to silicon-based materials can increase sensitivity in biosensing applications. Phage display identified two peptides with high affinity for porous silicon films, SPGLSLVSHMQT and LLADTTHHRPWT. When a biotin molecule was conjugated to SPGLSLVSHMQT, the detection limit for streptavidin was decreased by 21-fold [49].

#### 3.2.2. Platinum and Palladium

The physical and chemical properties of nanomaterials are regulated by their shape. A customizable, programmable crystal synthesis with predictable properties in nanomaterials can produce desired functions. Crystal growth is largely controlled by the surface energy of the facets where low-energy facets tend to persist and high-energy facets disappear resulting in shape enclosure by low-energy facets. Biomolecules such as peptides can be chosen for their affinity to specific facets, leading to a desired shape. Two peptides (TLTTLTN and SSFPQPN) bind specific platinum facets (Pt-{100} and Pt-{111}, respectively) and shift crystal formation toward nanocubes or nano-octahedrons, respectively [47]. Likewise, two peptides (LSNNNLR and SPSTHWK) have affinity for Pd-{100} [46].

Peptides can be controlled with external stimuli, such as pH, temperature, and light. Changes in pH can release peptides from a surface, allowing control of peptide density on the surface. When palladium nanocubes were capped with the LSNNNLR peptide at pH 7.5, the shape remained stable [46]; however, when pH was lowered to 2.2, the peptide was released leading to a new surface-energy profile allowing previously protected Pd atoms to be released and form a concave structure on the Pd-{100} facets [46]. This pH-dependent control of surface composition indicates that peptides might be used as a “smart-cap” for structural control of nanomaterials [46].

#### 3.2.3. Apatite

Hydroxyapatite (HA) is a versatile biomaterial that accelerates osteointegration and is used as a bone defect filling material. HA can also be designed to have osteoinductive properties via attachment of biological factors such as growth factors. Standard processing conditions (sintering at >1000 °C) cause biological factors to denature. Bone-like mineral (BLM), a biomimetic formation of carbonated apatite, provides an alternative to increase osteogenesis in vivo. Since BLM is formed under STP conditions, it is amenable to sequestering biological molecules [50]. The use of BLM in concert with exogenous factors can spatiotemporally regulate bone formation. Regarding surface engineering of HA and BLM, their chemistries cannot be readily modified to attach carboxyl, hydroxyl, or amine functional groups. Phage display identified several peptides, such as VTK [51], with binding specificity to BLM. Since these peptides show preferential adsorption to apatite present in a variety of tissues, including bone, dentin, and enamel, they can be used for tissue engineering by attaching a second peptide such as a growth factor to create a dual-functioning peptide to generate the desired effect as well as possibly in injury repair for tendon and ligament by creating a dual-functioning peptide adhesive with affinity for bone on one end and tendon/ligament on the other.

## 4. Phage Display for Dual-Functioning Peptides

A benefit of using peptides is the ease with which they can be combined. Many ECM proteins have multifunctional domains that work concurrently to allow cell adhesion as well as provide directions for other cell functions. Typically, bi-functional or dual-functioning peptides are equipped with one targeting peptide conjugated to another targeting peptide or a functional peptide. The fusion of two peptides to create a dual-functioning peptide can mimic the multifunctional domains of native proteins (Figure 6). Since load-bearing biomaterials are typically selected for their mechanical and chemical properties rather than their biointeractive properties, dual-functioning peptides provide a way to spatially control the function of adjacent cells or their integration into the surrounding tissue as well as provide biological or chemical cues.

Often, implanted devices do not have the molecular structure to control the surrounding cells to generate ideal integration in vivo. Metal implants, including those used in orthopedic procedures such as hip replacements, or cardiovascular therapies such as stents, are chosen due to their mechanical strength and durability as well as their inertness. While materials like titanium are often favored for implants, a lack of integration with the surrounding tissue can lead to negative outcomes such as restenosis [52]. A titanium-binding peptide identified via phage display and attached to the integrin-binding peptide, RGD, increases cell retention to the titanium surface [44]. Such a modular design was also used to create a PEGylated peptide, with four titanium-binding peptides, that prevents bacterial colonization and biofilm formation [53].

Polypyrole (PPy) is a conducting polymer with applications in drug delivery, nerve regeneration, and biosensors. Different dopants alter the conductivity or topography of PPy. One of these dopants is chlorine, which creates PPyCl. The T59 peptide (THRTSTLDYFVI) binds to PPyCl and stabilizes in serum [54]. When combined with RGD, T59-RGD increases adhesion to PC12 cells (a rat pheochromocytoma cell line) [54]. PPyCL-specific binding peptides attached to cell-binding peptides can be applied to uses in drug delivery vehicles and neural probe coatings.

Similar experiments have been done on polystyrene for drug-eluting stents that prevent restenosis (poly(styrene-*b*-isobutylene-*b*-styrene) [55]. The dual-functioning peptide FFSFFFPASAWGSSGSSGK(biotin)CRRETAWAC binds to polystyrene as well as endothelial cells and reduces platelet binding to the surface [55]. The inclusion of the peptide RRETAWA allows for binding of the α5β1 integrin present in endothelial cells and lacking in platelets [55].

Cell adhesion is an important objective in modifying biomaterials for tissue engineering purposes. Increasing the specificity of cell adhesion can increase the efficiency of cell-based therapies. When considering large tissue defects or clinically relevant applications of tissue engineering, promoting cell differentiation and the formation of functional ECM is important. The group responsible for the identification of the VTK apatite-binding peptide and the phosphorylated variants [51] also identified a peptide with affinity for human bone marrow stromal cells (DPIYALSWSGMA, DPI) [56]. These two peptides were combined to form DPI-VTK which exhibited greater binding affinity for apatite surfaces than VTK, pVTK, and VTK-RGD [56]. Additionally, DPI-VTK increased the adhesion strength of human bone marrow stromal cells (hBMSCs) and induced pluripotent stem cell (iPS)-derived MSCs with minimal adhesion of osteoblasts and fibroblasts compared with VTK-RGD [56]. Interestingly, the DPI domain was not only specific for MSCs, but also led to an increase in cell spreading, proliferation, and differentiation, all of which are necessary for tissue regeneration (Figure 7) [56].

## 5. Conclusions

Peptides derived from phage display can be used in a wide array of applications from imaging to tissue engineering to drug targeting. Phage display takes advantage of nature’s randomness to find a peptide that can provide a specific function without full understanding of the substrate structure. Phage display is used to identify peptides that: (1) target a particular cell type, (2) attach preferentially to an inorganic or organic synthetic material, and (3) elicit a precise cellular response. While awaiting the results from the multitude of studies about COVID-19, it may be useful to find a peptide that can specifically target the outer shell of the virus, a peptide that can interfere with some portion of its viral behavior, or both. This review demonstrates the usefulness of phage display and suggests new opportunities to take advantage of more than one type of phage display-derived peptides (i.e., peptides combining targeting–functional or targeting–targeting sequences) to engineer peptides with broader applications. As the number of targeting and functional peptides identified by phage display continues to grow, the number of studies focusing on dual-functioning peptides at the interface between biomaterials and tissue will continue to grow as well. Cell functions can be specifically and spatially controlled in a well-defined manner by using dual-functioning peptides. The modular design of most dual-functioning peptides lends itself to switching out the targeting or functional peptide based on the specific requirements of the material or desired cell type/function.

## Figures and Tables

**Figure 1 ijms-21-05994-f001:**
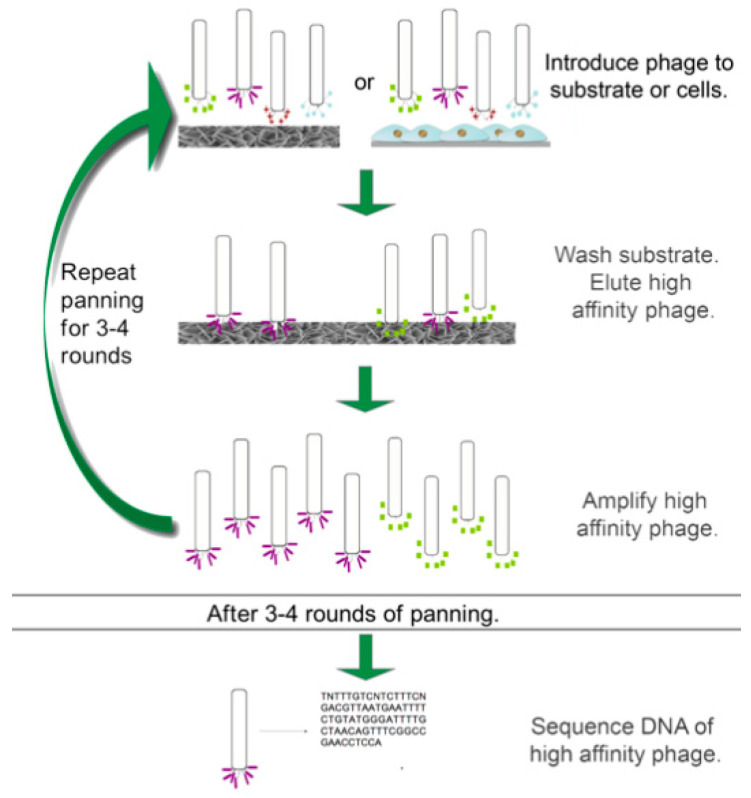
Schematic of panning stage of phage display. Phages are introduced to either a substrate or specific cell population followed by washing away non-adherent phages and eluting high affinity phages. High-affinity phages are amplified and the process is repeated at least 3 times to reach a consensus sequence. Image used with permission from Springer.

**Figure 2 ijms-21-05994-f002:**
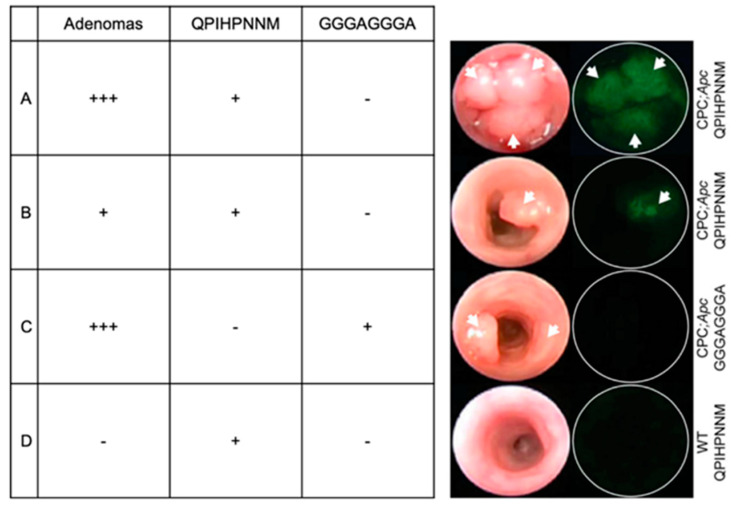
Wide-field endoscopy images acquired after in vivo incubation (5 min) of fluorescently labeled peptide QPIHPNNM in a genetic mouse model of adenoma formation in the colon. The left column shows white light images and the right column shows fluorescence images for binding of (**A**) QPIHPNNM to colon tissue with multiple adenomas, (**B**) QPIHPNNM to colon tissue with one adenoma, (**C**) control peptide GGGAGGGA to colon tissue with multiple adenomas, and (**D**) QPIHPNNM to lumen of colon tissue of a wild-type (WT) mouse. In the model shown, CPC:*Apc*, Cre recombinase is linked to the Cdx2 promoter and floxed *APC* (*adenomatous polyposis coli*). Images modified from [10] and used with permission from PLoS ONE.

**Figure 3 ijms-21-05994-f003:**
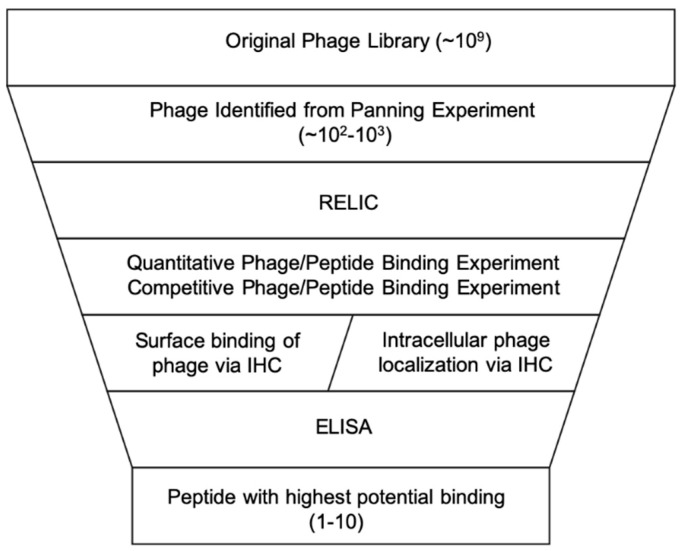
Representation of steps in the phage display process to narrow the phage library down to only peptides with high affinity to the target substrate or cell population. Image modified from [7] and used with permission from Springer.

**Figure 4 ijms-21-05994-f004:**
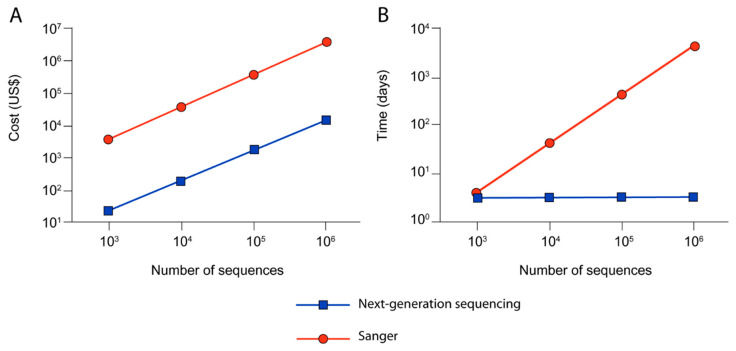
These graphs demonstrate how (**A**) cost and (**B**) time necessary for phage sequencing decreases when using next-generation sequencing (blue) over traditional Sanger sequencing (red) for increasing numbers of sequences. Images modified from [15] and used with permission from PLoS ONE.

**Figure 5 ijms-21-05994-f005:**
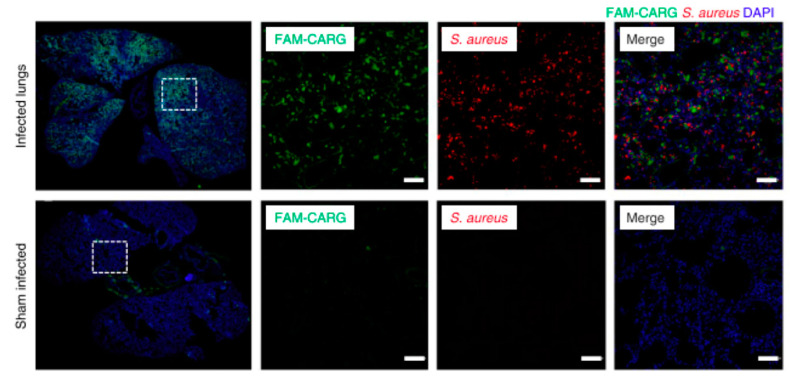
Representative immunofluorescence images of fluorescently labeled peptide CARG (FAM-CARG, green) homing and binding to lungs infected with *Staphylococcus aureus* (**red; top row**) and sham-infected lungs (**bottom row**) with DAPI (**blue**) staining of nuclei. In each row, the leftmost image shows lung tissue labelled with FAM-CARG (**green**) and DAPI (**blue**); the white box outlines the region of interest (ROI) examined at higher magnification (scale bar = 50um). In each row, the ROI is shown with individual fluorescent channels FAM-CARG (**middle left**) and S. aureus (**middle right**), and all fluorescent channels (**right**). FAM-CARG was intravenously injected into tails of infected mice and allowed to circulate for 30 min prior to sacrifice and histological examination (*n* = 3–5 mice per group with 5 histological samples examined per lung). Images modified from [43] and used with permission from Nature Biomedical Engineering.

**Figure 6 ijms-21-05994-f006:**
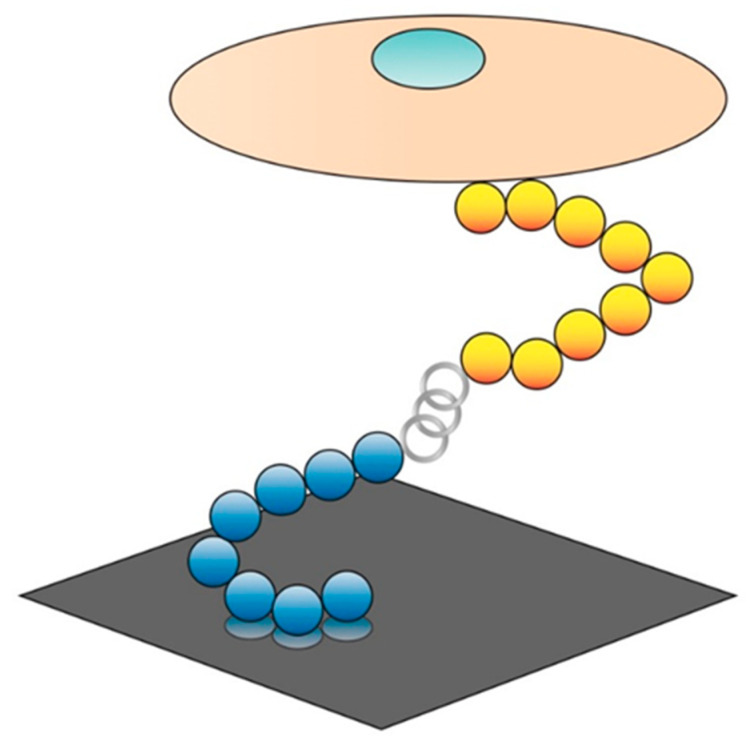
Dual-functioning peptide schematic with a targeting peptide (**blue**) attached to a substrate (**dark grey**) connected to a functional or specific cell targeting peptide (**yellow**) via spacer amino acids (**light grey**). Image modified from [44] and used with permission from Advanced Materials.

**Figure 7 ijms-21-05994-f007:**
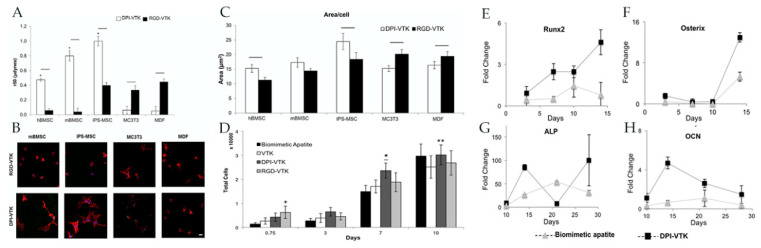
(**A**) Half-cell detachment force (τ50) measured from peptide-coated apatite films of human bone marrow stem cells, murine bone marrow stem cells (mBMSCs), induced pluripotent stem cell (iPS)-derived mesenchymal stem cells (MSCs), pre-osteoblastic MCTC3s, and murine dermal fibroblasts (MDFs). (**B**) Confocal microscopy images taken at 40x show increased spreading of mBMSCs and iPS-MSCs on DPI-VTK-coated apatite films and increased spreading of MCTC3 and MDFs on RGD-VTK-coated apatite films. (**C**) Quantitative analysis of cell spreading via ImageJ on peptide-coated apatite films. (**D**) Total number of cells over time, reaching saturation at day 10. * denotes significance compared to apatite control and tissue culture polystyrene (TCPS). * denotes significance compared to apatite control. ** denotes significance compared to VTK. (**E**–**H**) Relative gene expression of osteogenic markers (*Runx2, OSX, ALP,* and *OCN*) normalized to day 0 and *GAPDH* of iPS-MSCs on biomimetic apatite and peptide coated apatite. Images used with permission from Taylor & Francis.

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
