# Peer review of "Phage Display to Augment Biomaterial Function"

_ijms, 2020, doi:10.3390/ijms21175994_

Round 1

Reviewer 1 Report

This comprehensive review of phage display goes through the concept of phage display, procedures/methods of making and analyzing phage display libraries, including common terms of usage, a brief history and selection techniques. It then covers the broad fields for which phage display has been used. The comprehensive nature of this review makes it valuable to the field.

Major concerns:

*do the authors have the appropriate permissions to modify figures? a majority of figures in this manuscript are modified from other publications.

Minor comments:

*lines 53/54: phage are not commonly referred to as a molecule, and although the term molecule is used loosely (beyond covalently bond atoms), it should not be extended to something as complex as a phage. 

*Figure 1 could be improved.  For example, it is difficult to tell what phage are bound on  the black base that is shown.

*Figure 3 legend is missing a "["

*line 166 : miswording "by on"

*line 283: miswording "following phage display run"..

*line 354:  S. aureus should be italicized

Author Response

Major concerns:

Authors have been authorized/given rights and permissions to use all figures in the paper.

Minor concerns:

*lines 53/54: phage are not commonly referred to as a molecule, and although the term molecule is used loosely (beyond covalently bond atoms), it should not be extended to something as complex as a phage. 

  • Changed phrasing of lines 52-54 to more accurately describe the peptide expression present on the phage

*Figure 1 could be improved.  For example, it is difficult to tell what phage are bound on  the black base that is shown.

  • Added higher resolution version of Figure 1 that hopefully clarifies any confusion

*Figure 3 legend is missing a "["

  • Several “[“ had been accidentally deleted during the editing process; they have all been added

*line 166 : miswording "by on"

  • Deleted “on” now on line 189

*line 283: miswording "following phage display run"..

  • Deleted “run” now on line 313

*line 354:  S. aureus should be italicized

  • All instances of aureus have been italicized lines 411, 414, 415, 418

Reviewer 2 Report

The review describes the potential application of phage display technology to discover amino acid sequences with high affinity for a desired target. Specifically, the review deals: i) peptides with bind ability to protein, cell, tissue or inorganic material; ii) functional peptides able to modulate target’s activity after binding; iii) dual-functional peptides showing the combination of targeting and/or functional properties.

The review is quite organized, however in some section it almost completely misses of recently sentences/concepts with appropriate references. The review could be improved by including more details about:

  • The review focuses more on usefulness of phage display to discovery peptide able to target and/or act specific targets. Otherwise, the ability of the selected peptides to improve the biomaterial function is only mentioned but not described, as the title suggests.
  • I suggest to Authors to add recent references. For example, in the sections 1.2 and 1.3 the Authors described in vivo selection, followed of data processing of phage selected highlighting the advancement of the NGS. However, in the description are miss recently articles that describe in vivo selection of cancer tissues, also processed by NGS (see the following works: i) doi: 10.3390/v11110988; ii) doi: 10.1007/s00726-018-2539-1; doi:10.1039/c7cc09077c; doi: 10.1016/j.omto.2019.01.001).
  • 7 lines 235, the reference to the figure 3 in the text is incorrect.

Author Response

The review focuses more on usefulness of phage display to discovery peptide able to target and/or act specific targets. Otherwise, the ability of the selected peptides to improve the biomaterial function is only mentioned but not described, as the title suggests.

  • Added language about specific applications of phage display-derived peptides to alter the function of biomaterials
    • Lines 319-335: stem cell differentiation on biomaterial surface
    • Lines 354-358: promotion or inhibition of cellular function on biomaterials
    • Lines 379-381: imaging of pathological targets
    • Lines 392-394: microfluidic devices for beside diagnosis of disease
    • Lines 399-407; 425-427: tissue specific homing of drug delivery systems

I suggest to Authors to add recent references. For example, in the sections 1.2 and 1.3 the Authors described in vivo selection, followed of data processing of phage selected highlighting the advancement of the NGS. However, in the description are miss recently articles that describe in vivo selection of cancer tissues, also processed by NGS (see the following works: i) doi: 10.3390/v11110988; ii) doi: 10.1007/s00726-018-2539-1; doi:10.1039/c7cc09077c; doi: 10.1016/j.omto.2019.01.001)

  • The authors have read all and included some of the papers suggested by the reviewer referencing NGS
    • Lines 102-108 Wada et al, Mol Ther Oncolytics (2019);
    • Lines 149-155 Gillispie et al, Viruses (2019)
    • Lines 173-180 Liu et al, Amino Acids (2018)
  • Concerning the paper not included: while interesting, the authors felt inclusion may have taken the review on a tangent outside the scope of the topic.
    • The third paper listed would introduce another form of phage display that had not been discussed previously within the paper, but also did not add much to the paper without substantial additional explanation of the “bionanofibers”

7 lines 235, the reference to the figure 3 in the text is incorrect.

  • Reference to Figure 3 has been corrected; Line 263